# Critical Review for the Production of Antidiabetic Peptides by a Bibliometric Approach

**DOI:** 10.3390/nu14204275

**Published:** 2022-10-14

**Authors:** Ticiane Carvalho Farias, Thaiza Serrano Pinheiro de Souza, Ana Elizabeth Cavalcante Fai, Maria Gabriela Bello Koblitz

**Affiliations:** 1Graduate Program in Food Science and Nutrition, Federal University of the State of Rio de Janeiro (UNIRIO), Rio de Janeiro 22290-240, RJ, Brazil; 2Department of Basic and Experimental Nutrition, Institute of Nutrition, Rio de Janeiro State University (UERJ), Rio de Janeiro 20550-013, RJ, Brazil

**Keywords:** hypoglycemic bioactive peptides, protein hydrolysis, DPP-IV, α-amylase, α-glucosidase, inhibitory peptides

## Abstract

The current bibliometric review evaluated recent papers that researched dietary protein sources to generate antidiabetic bioactive peptides/hydrolysates for the management of diabetes. Scopus and PubMed databases were searched to extract bibliometric data and, after a systematic four-step process was performed to select the articles, 75 papers were included in this review. The countries of origin of the authors who published the most were China (67%); Ireland (59%); and Spain (37%). The journals that published most articles on the subject were Food Chemistry (n = 12); Food & Function (n = 8); and Food Research International (n = 6). The most used keywords were ‘bioactive peptides’ (occurrence 28) and ‘antidiabetic’ (occurrence 10). The most used enzymes were Alcalase^®^ (17%), Trypsin (17%), Pepsin, and Flavourzyme^®^ (15% each). It was found that different sources of protein have been used to generate dipeptidyl peptidase IV (DPP-IV), α-amylase, and α-glucosidase inhibitory peptides. In addition to antidiabetic properties, some articles (n = 30) carried out studies on multifunctional bioactive peptides, and the most cited were reported to have antioxidant and antihypertensive activities (n = 19 and 17, respectively). The present review intended to offer bibliometric data on the most recent research on the production of antidiabetic peptides from dietary proteins to those interested in their obtention to act as hypoglycemic functional ingredients. The studies available in this period, compiled, are not yet enough to point out the best strategies for the production of antidiabetic peptides from food proteins and a more systematic effort in this direction is necessary to allow a future scale-up for the production of these possible functional ingredients.

## 1. Introduction

Diabetes is an old disease of which there are at least three known types: type 1 usually occurs during childhood and adolescence, as a result of the failure of the β-cells in the pancreatic islets to secrete insulin, caused by an autoimmune condition; type 2 is the most common and is usually due to eating habits and lifestyle; and there is also gestational diabetes. It is estimated that type 2 diabetes (T2D) accounts for more than 90% of the total cases, thus, the research for strategies to prevent, cure, or slow down the progress of this disease is a subject of interest to the scientific community. In 2017, approximately 462 million individuals were affected by T2D, which corresponds to 6.28% of the world population, and this number is projected to increase to 700 million by 2045. It is estimated that this disease and its complications caused 4.2 million deaths globally in 2019, making it the ninth leading cause of mortality worldwide, and its prevalence has been growing faster in low- and middle-income countries than in their high-income counterparts. The fraction of the adult population with diabetes in 2019 in high-income countries was 10.4%, followed by middle-income countries with 9.5%, whereas low-income countries showed the lowest prevalence (4.0%). Prevalence is projected to increase in the three income groups to 11.9%, 11.8%, and 4.7%, respectively, by 2045.

Diabetes *mellitus* is a metabolic disorder that is a worldwide public health concern and poses significant economic and social challenges [1]. Research has been confirming that food protein hydrolysates with in vitro DPP-IV, α-amylase, and α-glucosidase inhibitory properties are potential agents against T2D [2]. Moreover, the American Diabetes Association and the European Association for the Study of Diabetes have approved the use of these hydrolysates as antihyperglycemic drugs [3]. Thus, the development of potent antidiabetic hydrolysates containing bioactive peptides with specific inhibitory activity is of great scientific interest [2].

Treatment against T2D consists of lifestyle modification and the use of drugs that increase glucose uptake in tissues, decrease gluconeogenesis, or stimulate insulin secretion. A relatively novel treatment involves dipeptidyl peptidase IV (DPP-IV), α-glucosidase, and/or α-amylase inhibitors that display a hypoglycemic effect by reducing intestinal glucose absorption and enhancing the synthesis of insulin, as well as acting as receptor agonists of glucagon-like peptide-1 (GLP-1) [1,4]. At present, there is no cure for diabetes mellitus and the use of synthetic drugs that may have side effects is a widely applied treatment. The hazards associated with these therapies include hypoglycemia, weight gain, tiredness, diarrhea, and anemia, among others [5]. Therefore, the development of natural food-derived peptide remedies may not only avoid the side effects, but also allow the early intervention and nutritional treatment of patients. Thus, they may be the ideal choice for preventing diabetes and improving its treatment [6]. To be effective, peptides need to be released from the protein matrix, be bioaccessible, and reach the active site in sufficient quantities to exhibit biological activity. Recently, several studies have been focusing on the generation of bioactive peptides from food proteins and their utilization as functional ingredients.

Literature reviews play an essential role in academic and technological research, gathering available knowledge, and pointing out the state of the art in a given field. The compilation of academic knowledge grows challenging, as scientific knowledge keeps increasing exponentially and thousands of new articles are published on a daily basis. This task becomes even more troublesome with the advent of predatory journals publishing open access articles without proper peer review. Bibliometric reviews can help to overcome several difficulties encountered in collecting reliable bibliographic data: separating what has already been tested from what still needs to be tested or better evaluated, relating methods used and the results obtained, and indicating what is known and what is not known in that specific field, in a quantitative way [7]. The goal of the present study was, therefore, to carry out a quantitative survey of the knowledge available in reliable databases about the factors involved in the production of peptides with antidiabetic activity in order to provide an overview of what has already been successfully tested, what needs to be further evaluated, and, by exclusion, what has not yet been studied in this topic.

## 2. Methodology

### 2.1. Search Strategy of the Bibliometric Analysis

The bibliometric review was based on the methodological research proposed by Randhawa, Wilden, and Hohberger [8]. The search approach was conducted using ‘Scopus’, a major citation database of peer-reviewed literature, and ‘PubMed’, an index of biomedical literature, including all data available in the ‘Medline’ database. The search included published papers (original and review papers) that had focused on protein hydrolysates containing bioactive peptides with antidiabetic properties that could be applied to functional foods and nutraceuticals.

The first step consisted in identifying the descriptors to define three cores: (i) the active principle, by including ‘bioactive peptide’ or ‘inhibitory peptide’, (ii) the type of inhibition, and (iii) the expected bioactivity. For the ‘Scopus’ database search, the following descriptors were used: (“bioactive peptide” OR “inhibitory peptide”) AND (“alpha glucosidase” OR “alpha amylase” OR “dipeptidyl peptidase” OR “DDP-IV”) AND (“antidiabetic” OR “anti-diabetic”). The same descriptors were applied for the ‘PubMed’ database search, however, without the use of quotes framing each keyword.

### 2.2. Review Process and Selection Criteria

In the next step, two authors, working independently, identified the relevance of the papers by reviewing the titles, abstracts, and keywords that appeared in each search. The publication period taken into consideration for the present review was between 2016 and May 2021, and the whole search and selection process was performed between 21 May to 28 May 2021. All the selected papers were reviewed to avoid any loss of crucial references. An article was included for the analysis only if after screening: (i) it was published between 2016 and 2021 (ii) it was a peer-reviewed research or review article; (iii) it was in the final publication stage; (iv) it was published in English; (v) and that there was a full text available. While the records excluded were (i) non-research articles (e.g., book chapter, short survey, documents); (ii) there was any missing data in the article (e.g., authors’ name); (iii) if it was not published, it means that articles in press were not considered; (iv) articles published in a different language of English (e.g., Chinese); (v) articles that the full text was not available (e.g., only the abstract available); (vi) non-peer-reviewed articles; (vii) and off-topic studies (e.g., those who did not make enzymatic hydrolysis to generate bioactive peptides).

### 2.3. Data Extraction

The data extracted for each individual study included the following: database that the article was identified; keywords; country of publication; year; journal of the publication; type of the article (e.g., original or review); inhibition type; multifunction peptides assays; enzyme used; source of protein used; type of analysis (e.g., in vitro, in vivo, or in silico); peptide sequence; outcomes; and challenges and perspectives.

## 3. Results and Discussion

This bibliometric review intended to provide a survey of recent articles that presented research on antidiabetic bioactive peptides from different dietary protein sources to be used as food ingredients and improve glycemic regulation of diabetic patients.

In Figure 1, a flow chart shows the records identified in each database and the total number of articles (n = 386); the records that were screened and the eligible ones; and the total number of studies included in this review (n = 75). After a refined search, 28 papers were found to be present in both databases. Of the total of 75 papers in the final selection, 22% (n = 17) were exclusive to the ‘Scopus’ database and 40% (n = 30) were exclusive to the ‘PubMed’ database.

### 3.1. Number of Publications, Authors’ Countries/Territories, Publication Period, and Leading Journals

Among the 75 included articles, original articles accounted for 85% (n = 64) of the selected articles during the whole period (2016 to May 2021), and they focused on different methods to obtain peptides from several matrices. Over the target period of publications, there was an increase in the number of identified peptide sequences, with a greater number of peptides reported for the first time in 2020 and 2021. This shows the growing interest in this subject and reinforces the usefulness of compilations such as the present study.

Among the 27 countries that studied antidiabetic peptides, there was a total of 98 researchers who authored the publications. The countries with the highest number of publishing authors were China (n = 18; 67%); followed by Ireland (n = 16; 59%); then Spain (n = 10; 37%). In terms of distribution of published articles per continent, Asia (n = 38; 39%) published the highest number of articles, followed by Europe (n = 34; 35%), the Americas (n = 14; 14%), Africa (n = 8; 8%), and Oceania (n = 4; 4%). Globally, China (116.4 million), India (77.0 million), the USA (31.0 million), Pakistan (19.4 million), Brazil (16.8 million), Mexico (12.8 million), Indonesia (10.7 million), Germany (9.5 million), Egypt (8.9 million), and Bangladesh (8.4 million) were the top 10 countries with the highest number of diabetics in 2019 [9]. These data may explain the interest shown by Chinese as well as other Asian researchers in this topic, but also points to the fact that academic research on antidiabetic peptides is not specifically located in countries with the highest incidence of this disease.

Three journals have published most of the papers about antidiabetic peptides. For these particular journals, there was a correlation between the year of publication and the authors’ countries of origin through the Sankey diagram (Figure 2). The journals that most published on subjects related to DPP-IV, α-amylase, and α-glucosidase inhibitory peptides were Food Chemistry (n = 12); followed by Food & Function (n = 8); and then Food Research International (n = 6). When the top publishing journals and the author’s countries were correlated, it was found that the country that appeared most frequently, Ireland (n = 12), corresponded for about half of the articles published in these leading journals (total number of published articles in the three leading journals = 26), compared to the United Arab Emirates and Spain (n = 4, each), China (n = 3), Malaysia and the USA (n = 2, each), while the other authors’ countries in the diagram had 1 published article each. Therefore, it appears that although China is the country of origin with the highest overall number of authors/publications, it did not contribute the most to the three main journals when publishing on the subject.

### 3.2. Author’s Keywords

The author’s keywords that appeared among the selected published articles more than twice, were recorded and are shown in Figure 3. Two hundred and eighteen different keywords were found among the included articles, of which 46 appeared at least in two different papers. The most used keywords were: ‘bioactive peptides’ (occurrence 28, total link strength 54) and ‘antidiabetic’ (occurrence 10, total link strength 38). When the keywords were grouped according to their meaning, the top five groups were: ‘bioactive peptide, bioactive peptides, and bioactivity’ (occurrence 38), which also showed up as the main keyword-group followed by ‘DPP-IV’ and its variations, ‘dipeptidyl peptidase IV, dipeptidyl peptidase IV inhibition, dipeptidyl peptidase-IV and DPP-IV inhibition’ (occurrence 25); ‘antidiabetic, antidiabetic activity and antidiabetic potential’ (occurrence 16); ‘α-glucosidase, α-glucosidase inhibition, α-glucosidase inhibitory activity’ (occurrence 10); and ‘protein hydrolysate, and protein hydrolysates’ (occurrence 9). As expected, the most cited keywords match the descriptors used in the bibliometric search. The evaluation of the keywords highlighted that, within the most tested enzymes related to diabetes, DPP-IV was the most represented in the author’s keywords, indicating that it has been the most tested, followed by alpha-glucosidase, the second most tested enzyme.

Other groups of keywords were ‘antioxidant, and antioxidant activity’ (occurrence 11); and ‘ACE, ACE inhibition, and angiotensin-converting enzyme’ (occurrence 7). These latter keywords appeared very frequently, owing to the multifunctionality of many bioactive peptides, which, in addition to acting as antidiabetic agents, may present other bioactivities, e.g., antioxidant and hypotensive capacities. Other keywords also frequently appeared among the selected articles and their occurrence was mostly related to peptide bioactivities; the methods used to identify these peptides; the diseases that might be suppressed/treated by their use; and the proteins used as a substrate to obtain the hydrolysates.

### 3.3. The Most Used Proteases

Bioactive peptides are inactive in the sequence of the native protein molecule; however, they may be released through protein hydrolysis [10]. This reaction is based on the cleavage of the peptide bonds, which generates peptides of different sizes. Depending on the aminoacidic sequence of these peptides, the hydrolysate might present some biological activity [11]. To produce hydrolysates, protein hydrolysis can be carried out by methods such as chemical treatment; microbial fermentation; or enzymatic hydrolysis, by using proteases. Among these techniques, hydrolysis by proteases is the most frequently used to produce bioactive peptides [12]. Some of the advantages of enzymatic hydrolysis, when compared to chemical hydrolysis or fermentation processes, are: the generation of final products free from toxic compounds and organic solvents; an improved reaction rate; also, this technique is performed under mild temperature and pH conditions; enzymes present high specificity and good stereoselectivity; the reaction normally does not produce secondary products; the use of protease provides an environmentally friendly procedure; enzymatic hydrolysis usually produces higher peptide yields, which also have good quality; enzymes are easy to inactivate; and in general, the process is fairly simple [12,13]. As disadvantages, several parameters, such as enzyme-substrate ratio, hydrolysis time, pH, and reaction temperature, must be monitored and managed in an optimal range during enzymatic hydrolysis, because these factors may affect diverse properties in the resulting peptides. Usually, a pre-treatment of the substrate is necessary to enhance hydrolysis; and, last but not least, this method presents a higher cost, in comparison to other processes such as fermentation, owing to the high price of the enzymes [14].

Twenty-one different enzymes were identified among the selected published articles. The most used enzymes were Alcalase**^®^**, also referred to as subtilisin (EC 3.4.21.62) (n = 23; 17%), followed by trypsin (EC 3.4.21.4) (n = 22; 17%), pepsin (EC 3.4.23.1) (n = 20; 15%, each), and Flavourzyme**^®^** (a commercial mixture of fugal endo- and exo-peptidases (EC 3.4.11.1)) (n = 15; 11%).

Alcalase**^®^** is the trade name of the subtilisin produced by the Novozymes company. It is a microbial peptidase, produced from *Bacillus licheniformis* [15]. This protease is a serine S8 endo-proteinase that presents broad specificity and prefers to cleave large uncharged residues in the P1 position [13]. Owing to its broad specificity, this enzyme usually generates peptides with low molecular weight [16], which may explain its frequent use, as many peptides that present a variety of bioactivities, including antidiabetic activity, also show low molecular mass [16,17].

Trypsin is an animal protease, and all the studies included in the present review used trypsin extracted from porcine pancreas. This enzyme is an endopeptidase that shows specificity to cleave bonds involving arginine and lysine residues [18]. Its narrow specificity usually leads to the release of peptides with bioactivities owing to their terminal residues of R or K, which are considered molecular features for peptides that can inhibit α-glucosidases [19]. Trypsin has been used to obtain protein hydrolysates with different industrial uses for some time and presents the advantage of producing peptides possibly resistant to digestion by pancreatic enzymes.

Pepsin used in all papers included here was obtained from porcine gastric mucosa [20]. This protease presents broad specificity and prefers to cleave peptide bonds involving aromatic and hydrophobic amino acids [18]. According to Gomez et al. [21], this ability to break aromatic amino acid linkages might be related to increased DPP-IV inhibitory activity, as DPP-IV inhibitory peptides generally show a branched-chain amino acid or an aromatic residue containing a polar group in the side chain at their N-terminal position and/or a proline residue located at P1. Like trypsin, pepsin has several traditional industrial applications and presents the possibility of generating peptides resistant to gastric digestion.

Flavourzyme**^®^** is a trademark name given to a mixture of proteolytic enzymes by the Novozymes company. Flavourzyme**^®^** is produced by *Aspergillus niger* and exhibits endo- and exopeptidase activities [22]. Owing to its exopeptidase activity, Flavourzyme**^®^** is able to release very small peptides and free amino acids [18], which may favor the generation of hypoglycemic peptides, as DPP-IV and α-glucosidase inhibitory activities have been associated with low molecular peptides [16,17]. Exopeptidase activity also exerts a debittering effect, which helps to produce more palatable hydrolysates.

The other commonly used enzymes in the studies were Papain (EC 3.4.22.2) (n = 9; 7%); followed by Pancreatin (EC 232-468-9) or Corolase PP**^®^** (n = 10, 8%); Protamex**^®^**, Neutrase**^®^**, and Chymotrypsin (EC 3.4.21.1) (n = 6; 5% each); Bromelain, or Bromelin (EC 3.4.21.33) (n = 4; 3% each); Thermolysin**^®^** (EC 3.4.24.27), and Pronase E**^®^** (n = 2; 2% each). Protamex**^®^** (from *Bacillus licheniformis*, comprising a mixture of endo- and exopeptidases), Neutrase**^®^** (endopeptidases from *Bacillus amyloliquefaciens*), Thermolysin**^®^** (endopeptidases from *Geobacillus stearothermophilus*), Pronase E**^®^** (from *Streptomyces griseus*, containing both exo- and endopeptidases) are microbial proteases [15,23,24,25] whereas pancreatin (from porcine pancreas), chymotrypsin (from bovine pancreas) [25,26], and Corolase PP**^®^** [27] are digestive enzymes extracted from animal sources. Papain (papaya, *Carica papaya*), and bromelain (pineapple stem, *Ananas comosus*), in turn, are commercially available plant proteases [21,25]. Other proteases that appeared in just one study were alkaline proteinase, an animal proteolytic enzyme, PROTIN SD-NY10**^®^** (*Bacillus* metalloendopeptidase—EC 3.4.24.28), zingipain (ginger, *Zingiber officinale* EC 3.4.22.67), Protease from *Streptomyces griseus*, Proteinase K (EC 3.4.21.64), and Thermoase.

The proteolytic enzyme or set of enzymes used to obtain hydrolysates and peptides is possibly the single most important tool in the generation of bioactive peptides from food proteins. The peptides encrypted in the amino acid sequence of the proteins will be released in their active form according to the specificity of the applied enzyme. Likewise, the ability to generate peptides that resist digestion or that generate active derivatives after the action of digestive enzymes also depends on the mode of action of the enzymes applied in the production of hydrolysates. Even considering only enzymes readily available on the market, at affordable prices, and in considerable quantities, the number of studies exploring the different possibilities of obtaining bioactive peptides in general and antidiabetic peptides, in particular, is still very small. There is a wide range of possibilities to be explored, including the systematic evaluation of the application of several different enzymes on the same protein substrate and the systematic study of the same enzyme used on several different protein sources.

### 3.4. Main Protein Substrate Sources 

Peptides with bioactive functions can be obtained from various protein sources, and the most common are animal and plant dietary proteins. Protein substrates differ mainly in their amino acid sequences and their structural complexities. Likewise, bioactivity, environmental, social, and economic factors must be considered for generating hypoglycemic peptides. For this study, among animal proteins, fish, chicken, pig, marine animals, and milk provided by different species of mammals were included. Among the proteins of plant origin, grains, oilseeds, and seeds of different vegetables and fruits were included. In addition to the conventional sources, non-conventional protein sources were also considered, as shown in Figure 4. These are unusual foods in the diet of most of the population so far, but they may be good sources of various nutrients, such as proteins and derived peptides.

Among the protein substrates for obtaining antidiabetic peptides, until 2020, according to Rivero-Pino and co-workers [18], the most studied were milk and soy proteins, owing to their high biological value when compared to other proteins. Of the selected articles, 37% (n = 28) used substrates of animal origin, of which cow’s milk proteins (occurrence 9) were the most tested, not only for antidiabetic activity, but for several others as well. However, cows were not the only milk sources; camel’s milk was also present (occurrence 7); as well as donkey’s (occurrence 1) and mare’s (occurrence 1). According to Akan [28] knowledge of the DPP-IV inhibitory activities of milk from non-bovine mammals is quite limited, and the α-glucosidase inhibitory activity from donkey milk is utterly unknown. Other animal protein sources were also studied: fish (occurrence 5), chicken (occurrence 2), pork (occurrence 1), and marine animals (occurrence 4), e.g., Antarctic krill, which is an important marine organism for sustaining the food chain in the Antarctic Ocean ecosystem as feed, and a source of all essential amino acids [29,30].

Plant protein sources were leaves, seeds, and fruits. Fruit seeds are considered an economical source of proteins and the limited knowledge about these proteins shows that this field still needs to be explored to elucidate questions about the potential bioactivity of these compounds both in vitro and in vivo [5]. Twenty-eight articles (37%) dealing with proteins of plant origin were selected, among which beans were the protein source that appeared in the largest number of articles (occurrence 5), followed by soybeans (occurrence 4) that, when compared to milk proteins, presented higher α-glucosidase inhibition capacity. Soy hydrolysates also presented the highest inhibition of DPP-IV when compared with other plant sources such as lupine and quinoa [13,19]. Common beans are widely consumed pulses all over the world owing to their high nutritional and nutraceutical values. Among pulses, common beans appear as a promising alternative to obtain bioactive naturally occurring, and also encrypted peptides with antidiabetic activity. The first report of potential peptides of common beans to inhibit DPP-IV and α-glucosidase enzymes appeared in 2016 and was proven in 2019 by in vitro and in vivo studies [31,32]. Other plant protein sources such as rice bran, quinoa, walnut, and wheat gluten had two occurrences each and, with one occurrence each, corn germ peptides, cumin seed, orange seed, perilla seed, rape napin, kiwicha, camellia seed, pigeon pea, and barley bran were selected. A study on oilseed proteins and a comparison between vegetables (potatoes, sweet potatoes, yams, taro) were selected as well.

For the present survey, six articles from non-conventional protein sources were selected, among which four were about insects, one dealt with algae, and one was a review on peptides with DPP-IV inhibiting activity from marine organisms, including macro- and micro-algae, marine sponges, fish skin gelatin, and even tuna juice hydrolysates. In this work, the authors suggest that the valuable antidiabetic activities associated with bioactive peptides, derived from marine organisms and their metabolites, may be applied in the future by the nutraceutical and pharmaceutical industries with good results in the fight against diabetes [18,33]. Of the selected reviews, seven carried out studies with protein sources from several origins, three were focused on plant proteins, and one, as mentioned above, was based on marine sources.

Edible insects are among the unconventional food sources that, since 1 January 2018, are considered novel foods by European countries (European Council Regulations No. 2015/2283, 2015). Insects have been gaining more and more space owing to the need to defocus from animal and plant protein sources, overloaded by environmental issues, and also because of their high protein content. Crickets, for instance, show approximately 62% protein (dry matter) in their composition [34]. Research with edible insect peptides is at an early stage, but with promising results so far. Given the lack of studies on DPP-IV inhibition activity, the comparison is difficult; however, the inhibition capacity of these hydrolysates ranged from 62–69%, especially with low molecular mass peptides (<1 kDa ~10 amino acids), which was considered a satisfactory rate by the authors, encouraging future in vivo tests [34,35]. In recent work with edible earthworms, Rivero-Pino (2020a) performed an ultrasound pre-treatment and managed to increase the percentage of α-glucosidase inhibition of the hydrolysate, opening up another research front for the preparation of new peptides.

As important as the choice of enzymes is the selection of the protein substrate for the production of antidiabetic peptides. This protein must be edible and non-toxic and must have peptides with antidiabetic activity encrypted in its amino acid sequence. The following characteristics are still desirable: low cost and low impact on the environment. An interesting alternative could be the use of proteins from agro-industry residues and coproducts, which may help to lower residue generation, decrease waste, and contribute to the circular economy.

### 3.5. Bioactive Peptides with Multifunctional Activities

As shown in Figure 5, some of the published papers carried out studies that evaluated bioactivities other than antidiabetic capacity, focusing on the search for multifunctional peptides that act on various mechanisms of the consumers’ welfare. The antioxidant and antihypertensive capacities were the most widely studied. Works that tested enzyme inhibition, such as α-amylase, α-glucosidase, or DPP-IV, to access only the hypoglycemic potential of peptides, representing 60% (n = 45) of all selected papers. Studies that evaluated two and three peptide functions, including antidiabetic activity, represented 21% (n = 16) and 15% (n = 11), respectively. The vast majority of the latter (10 out of 11 occurrences) tested antidiabetic, antioxidant, and antihypertensive capacity, whereas the other tested antidiabetic, antioxidant, and immunomodulatory capacity [34]. 

In several selected studies, multifunctional peptides were found; however, not all fractions exhibited dual or multiple bioactivities. Studies that performed peptide identification by mass spectrometry were able to identify multifunctional peptides more clearly, such as the case of the peptide KTYGL, from bean [36]; KVEPLP and PAL, from Antarctic krill; QHPHGLGALCAAPPST, from quinoa [37]; the di-peptides LN, VE, and IP also from bean. These are examples that presented the most potent simultaneous DPP-IV and ACE inhibition [15]. Free amino acids, from Atlantic salmon, such as Y, F, W, and P, also showed multifunction as ACE and DPP-IV inhibition, and antioxidant activity [27]. In studies with synthetic peptides, Ibrahim et al. [38] identified that peptides STYV and STY were able to inhibit α-glucosidase and lipid accumulation in differentiated 3T3-L1 adipocytes.

Antioxidant activity was present in 19 works. This activity is usually attributed to low molecular weight peptides containing hydrophobic amino acids such as A, V, I, L, F, Y, P, M, and C, which increase their solubility in hydrophobic or lipidic phases. These hydrophobic amino acids are likely to interact with the S1 hydrophobic subsite of the DPP-IV active site [17,39,40] thus, they also act as hypoglycemic peptides. The presence of amino acids with hydroxyl groups (S, W, and Y) or basic amino acids (K and R) at the amino-terminus of peptides may play a critical role in α-glucosidase inhibition. Recent studies indicate that lower molecular mass peptides can also be responsible for the inhibitory activity of α-glucosidase, α-amylase, and DPP-IV [24,32,34,41]. In a study by Akan et al. [28], the correlation between the antioxidant activity by the ABTS (2,2′-azino-bis(3-ethylbenzothiazoline-6-sulfonic acid) method and the inhibition of DPP-IV showed a coefficient of 0.97. The presence of hydrophobic peptides in the sequence may also be linked to the inhibition of the pancreatic lipase, delaying the digestion/absorption of triglycerides [37].

Seventeen works tested peptides for anti-hypertensive activity together with antidiabetic activity. This activity is evaluated on the basis of the inhibition of the angiotensin I converting enzyme (ACE), which is a central component of the renin-angiotensin system, a potent vasoconstrictor that works by converting angiotensin I into angiotensin II, thus increasing blood pressure. Most of the purified DPP-IV inhibitory peptides reported were also di-, tri-, and oligopeptides. Dipeptides, such as IW, WL, and VY, naturally found in milk proteins and said to be antihypertensives, have also been identified in soy and rice proteins, showing that plant and animal proteins may be comparable sources of antidiabetic peptides. These smaller peptides can resist gastrointestinal digestion; thus, they may be absorbed into the circulatory system in an intact form [13,21]. The degree of hydrolysis plays an important role as most dietary protein derived peptides with ACE inhibitory abilities show relatively low molecular masses, generally ranging from dipeptides to pentapeptides, with molecular mass of 150–800 Da, and they contain positively charged K or R at the C-terminal end of sequences, which shows that the activity of a peptide may vary depending on the type of amino acid and their position in the sequence. Peptides containing amino acids such as Y, F, W, and P may be strongly linked to ACE inhibition [21,27,34,40,41]. Other works dealing with multifunctions complementing antidiabetic activity were also selected; peptides from different sources were tested in vivo or in silico, as shown in Figure 5.

One of the most interesting features of bioactive peptides is their multifunctional potential. Thus, the same peptide may present several beneficial effects and, even if each individual effect is not very high, the combination of different bioactivities contributes to an overall desirable result.

### 3.6. Type of Inhibition, Type of Analysis, and Type of Document

Table 1 shows that 32% of the articles (n = 24) were able to identify at least one sequence of peptides with a regular response function to diabetes, whether in vitro or *in silico*. This shows that omics and bioinformatics tools are extremely important to correctly identify and predict the probability of a given peptide to exhibit some bioactivity, based on its amino acid composition and structure. The results for both, animal and plant peptides, as well as from other protein sources proved to be quite promising in this line of study, indicating good results even when compared to widely used synthetic inhibitors; the other articles are described in Appendix A (Appendix A).

In silico analyses using bioinformatics tools totaled seven articles (9%) in the 2019–2021 range. On the other hand, in vitro studies totaled about 42% (n = 32) in the same period. Regarding the in vivo analysis, only three articles (4%) performed tests in rats, two with peptides of plant origin (hydrolyzed common beans and walnut peptides) [32,56], and the other with peptides of animal origin (hydrolyzed chicken feet) [57]. Rivero-Pino et al. and Acquah et al., both in 2020, reviewed the production and functionality of peptides that were tested in vivo.

Appendix A (Appendix A) shows the type of inhibition, type of analysis, type of document, and citation count of the 75 selected articles. Most of the articles (n = 39, 53%) tested the ability of the peptides to inhibit just DPP-IV, followed by articles that tested the inhibition against all antidiabetic-related enzymes—α-glucosidase, α-amylase, DPP-IV (n = 11, 15%). Two review articles did not provide information about the study inhibitions. In general, the most tested inhibition capacity was against DPP-IV (n = 59), followed by inhibition against α-glucosidase (n = 31) and α-amylase (n = 23).

DPP-IV is a metabolic enzyme that is expressed in several tissues, such as kidney, liver, and intestinal brush-border membrane, which cleaves dipeptides from the N-terminus of polypeptides, in which proline or alanine is at the penultimate position (Taga et al., 2017). This enzyme regulates the degradation of glucagon-like peptide-1 (GLP-1) and glucose-dependent insulinotropic polypeptide (GIP) incretin hormones according to physiological needs [11,58]. GLP-1 and GIP are hormonal peptides released after the ingestion of food, which regulate blood glucose levels by stimulating insulin secretion from the pancreas in a glucose-dependent manner and inhibit glucagon release by pancreatic cells [53,59]. However, DPP-IV quickly degrades them, leading to a very short half-life of those incretin hormones. Thus, DPP-IV inhibitors can extend GLP-1 and GIP action, and also enhance blood glucose regulation, which is considered a novel treatment against T2D [58]. DPP-IV activity has been related to metabolic syndrome (MS) in obese children and adolescents by El-Alameey et al. [60], where serum DPP-IV activity of obese patients with MS was significantly elevated in relation to obese patients without MS and control subjects. Trzaskalski et al. [61] also attributed to DDP-IV an important role in metabolic disease. In this instance, however, the authors indicated a direct protein–protein interaction between DPP-IV and different ligands such as adenosine deaminase (ADA), caveolin-1,35 extracellular matrix (collagen and fibronectin), and C-X-C chemokine receptor 4 (CXCR4) as responsible for the inflammation process both in obesity and T2D. In this case, the simple inhibition of the catalytic activity of the enzyme would not prevent the installation or progression of MS. 

Alpha-glucosidase is a membrane-bound enzyme present in the epithelial mucosa of the small intestine (brush border of the enterocytes). This enzyme hydrolyses 1,4-α-glycosidic linkages present in oligosaccharides and then converts these long-chain carbohydrates into monosaccharides that are absorbable from the intestine into the blood. By inhibiting α-glucosidase, the digestion of complex carbohydrates is retarded; therefore, the overall absorption of glucose in the blood is delayed and hyperglycemia is prevented [59]. Alpha-amylase hydrolyzes complex carbohydrates, such as starch, into oligosaccharides, which may be further hydrolyzed by α-glucosidase. This enzyme is secreted by the salivary and pancreatic glands [18]. The inhibition of α-amylase may lead to an overall decrease in blood glucose levels, as this enzyme is among the leading enzymes of carbohydrate digestion in the body [59].

Eighty-five percent of the articles in this survey were original articles (n = 64) and the other fifteen percent were review articles (n = 11). Most of the 75 selected articles performed in vitro studies, only two articles did not mention the type of analysis performed, which were review studies. Of the 64 original works, 42 performed only in vitro assays, 15 tested both, in vitro and in silico, 2 in vitro and in vivo, 1 in vitro and in situ, and 1 in vitro, in vivo, and in situ, and 3 performed only in silico studies. Among the review articles, most provided an overview of studies that reported in vitro and in vivo (n = 4); in silico and in vivo (n = 2); in vitro and in silico (n = 1); and in vitro (n = 1) anti-diabetic studies and one review paper included all three types of antidiabetic evaluations. These reviews discussed antidiabetic effects of protein hydrolysates and peptides generated from cereals and other types of grains, marine organisms, plants, and multiple protein sources. In addition, they showed recent advances in the subjects of how antidiabetic peptides are produced, as well as their properties, and the challenges of testing bioactive antidiabetic peptides in vivo.

The most cited articles were by authors Vilcacundo et al. [26], with 67 citations; followed by Uraipong and Zhao [62], with 61 citations; and Mudgil et al. [63], with 57 citations. These are original articles with in vitro tests. Vilcacundo et al. [26] evaluated quinoa protein, a plant protein, as well as Uraipong and Zhao [62], who studied rice bran proteins, while Mudgil et al. [63] tested hydrolysates of animal origin: camel milk proteins. The first two studies evaluated the multifunction of the peptides, such as antidiabetic and antiobesity; and antidiabetic and antihypertensive, respectively, while the last one evaluated only antidiabetic inhibition. As for the outcomes, the three studies concluded that the peptides generated from enzymatic hydrolysis showed the potential to be used as functional foods or for nutraceutical applications for the control of diabetes.

In most recent investigations, the inhibitory activities of enzymes by peptides are generally evaluated by in silico or in vitro assays, and most authors agree that in vivo experiments involving activity assays are needed to demonstrate the physiological effect of the peptides. However, the evaluation of the in vivo inhibitory activity of peptides is recommended to be performed after screening for inhibitory activity in vitro, as the in vivo analyses require longer times to study the effects in animals, usually between 4 to 6 weeks, in contrast to in vitro enzyme inhibition tests, which take only a few hours. Of the selected articles, eight mention in vivo analyses, five of them are reviews on antidiabetic peptides, two with in vivo analyzes using laboratory rats [32,57], and one with in vitro analyzes on cells and also in vivo analyses on rats [56].

According to Aquac et al. [1], the inhibition pathways studied so far used by the peptides are: inhibition of carbohydrases such as alpha-amylases and glycosidases; stimulation of incretin hormone production; stimulation of hormones that control satiety and gastrointestinal emptying and inhibition of DPP-IV. Recent studies involving the regulatory mechanisms of glucose metabolism by protein hydrolysates and peptides in general are relatively limited. There is a lack of scientific evidence on the molecular mechanisms directly related to the regulation of glucose metabolism at the cellular level. Studies have reported that proteins and peptides can regulate glucose metabolism by inhibiting DPP-IV activity in Caco-2 cells, stimulating GLP-1 secretion in GLUTag cells, and promoting glucose uptake in insulin-resistant HepG2 cells [6].

Wang et al. [56] studied the in vivo inhibitory activity of walnut hydrolysates with molecular weight of 3–10 kDa due to their higher alpha-glucosidase inhibitory activity (61.73%) and their ability to facilitate glucose uptake in insulin resistant HepG2 cells in vitro; and using rats with streptozotocin-induced diabetes (STZ). After six weeks of oral administration of the hydrolysates, the peptides alleviated insulin resistance by increasing its secretion and the levels of glucokinase and glycogen in the liver, as well as decreasing the blood glucose level in fasting of diabetic rats.

The same effect of decreasing glucose levels was observed in a study with peptides obtained from beans, which were administered orally to normoglycemic fasting rats and measured by starch tolerance tests after 45 min of starch ingestion [32]. Casanova-Marti et al. [59] showed that peptides from chicken feet reduced blood glucose in glucose intolerant rats, which may be partially due to their DPP-IV inhibitory activity, and also stimulated endogenous GLP-1 secretion, possibly involved in the antihyperglycemic action of this hydrolysate.

## 4. Challenges and Perspectives

The interest in studying peptides with bioactive potential has been growing in recent years, and this field has become very promising in the food and nutrition sciences. With the advance in science and technology, more studies are emerging from several protein sources, leading to the discovery of novel peptides, which may be used as natural food-derived peptides remedies, avoiding the usual side effects of the existing drugs. Thus, researchers have been exploring the potential of bioactive peptides as a new class of therapeutic drugs.

To archive success and acceptability when using a therapeutic agent, its delivery needs to be evaluated. The main form of drug administration is the oral route; thus, an oral active therapeutic agent must be stable in the gastric microenvironment. Even though peptide-based drugs are specific to their target and effective, oral bioavailability is still an issue that restricts the acceptance of this kind of drug and its market value [5].

As challenges, some of the articles addressed the investigation of the stability of peptide-based drugs under simulated gastric and intestinal conditions, as well as the bioavailability, metabolism, excretion, stability, and toxicity profile of the peptides. Owing to the extensive hydrolysis of peptides in the gastrointestinal tract by peptidases in the stomach, small intestinal brush border, and low cellular uptake of these peptides, the hydrolyzed peptides usually present very low oral bioavailability [64].

Most studies in this paper predict, as the next steps in research with antidiabetic peptides, the application of in vivo assays to unravel their mechanisms of action in live models, using animal models and human tests; research on the structure of the peptides to confirm their sequences, in different ways, such as isolation and sequencing; investigation of the relationship between the structural characteristics of peptides and their hypoglycemic activities; and the investigation of optimal strategies to scale-up the production, as well as the industrial optimization of the enzymatic production of bioactive antidiabetic peptides. Many are said to hold promise for replacing drugs in the future or serving as efficient functional food in preventing diabetes. For the case of peptides that were described for the first time in the last five years, more *in vitro*, in vivo studies, and cell culture assays are also needed to validate and confirm their functionality, and to provide a higher level of evidence.

## 5. Conclusions

A bibliometric analysis was carried out to identify research on antidiabetic bioactive peptides to be used as food ingredients and nutraceuticals. From what could be found in the available literature of the last 5 years, several studies have been conducted from different sources of food proteins, novel or traditional, with different commercial enzymes in order to obtain antidiabetic peptides. However, the compilations of these works still do not allow us to truly determine which types of proteins, of enzymes, and under which conditions the best peptides, or the best yield of peptides, with that specific function, are obtained. It can therefore be concluded that a more systematic investigation, possibly with the aid of bioinformatics tools, is desirable to help unravel the best approach for further upscaling of antidiabetic peptides, or any kind of bioactive peptides, production from dietary proteins. It is hoped that the compilation of findings in this paper will help researchers in this field to optimize strategies to point out for the best means of production of antidiabetic bioactive peptides to help improve the management and control of diabetes worldwide. 

## Figures and Tables

**Figure 1 nutrients-14-04275-f001:**
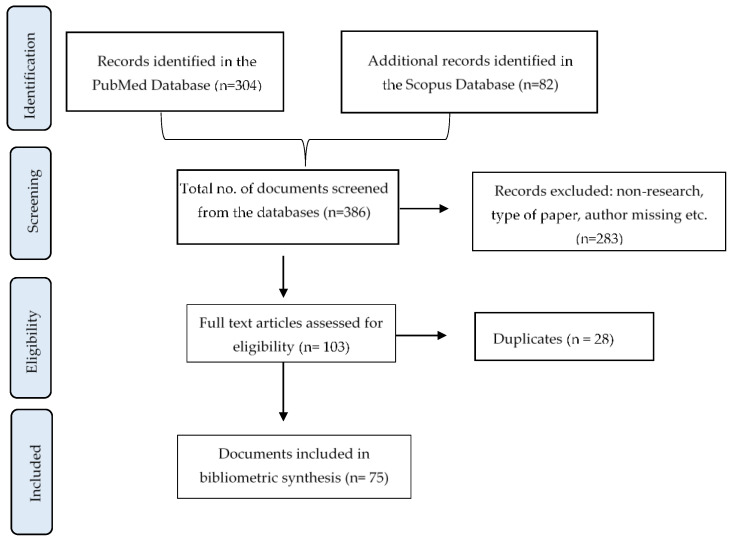
Flowchart indicating included and excluded articles.

**Figure 2 nutrients-14-04275-f002:**
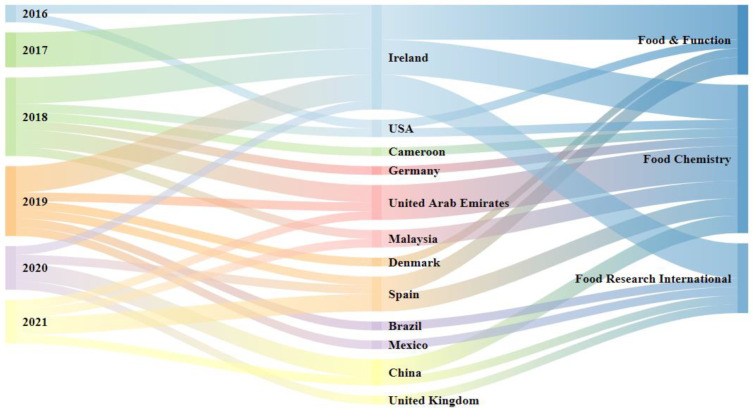
Sankey diagram for the selected articles relating year of publication, authors’ country of origin, and the three leading scientific journals that published the antidiabetic bioactive peptide studies.

**Figure 3 nutrients-14-04275-f003:**
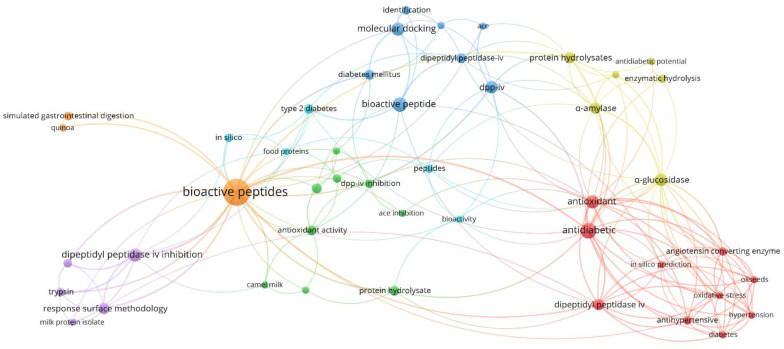
Network visualization map of the articles’ keywords that appeared at least twice. Node size is proportional to the number of occurrences (VOSviewer).

**Figure 4 nutrients-14-04275-f004:**
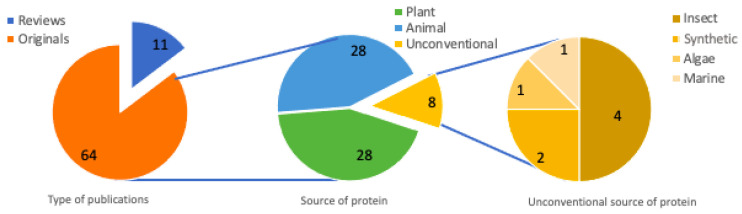
Source of the proteins used in the selected studies.

**Figure 5 nutrients-14-04275-f005:**
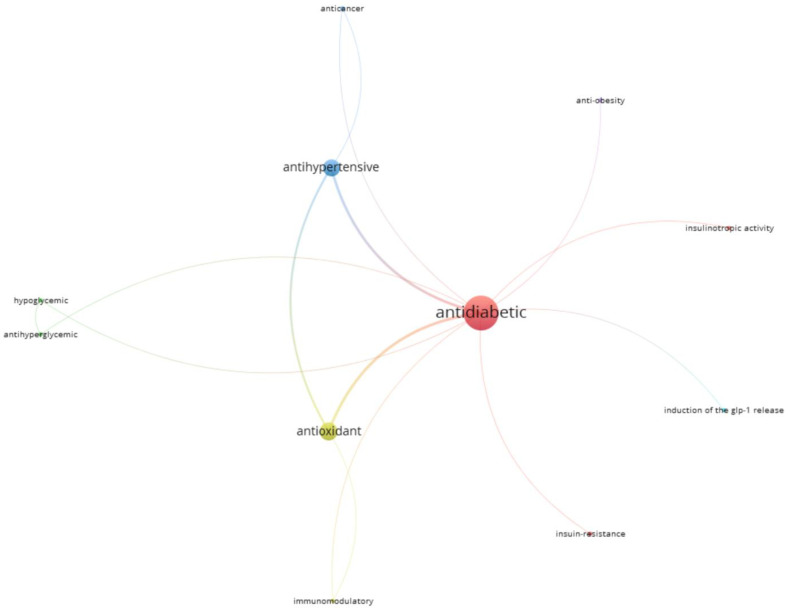
Network visualization map of the studied bioactive peptides with multifunctional activities. Node size is proportional to the number of occurrences (VOSviewer).

**Table 1 nutrients-14-04275-t001:** Selected studies with peptide sequence and their respective outcomes of interest.

Authors (Year)	Source of Protein	Peptide Sequence	Outcomes of Interest
Mudgil, et al., [42]	Bovine milk (*Holstein Friesian*) and dromedary camel milk (*Camelus dromedarius*, local breed)	FLWPEYGAL; LPTGWLM, MFE and GPAHCLL and HLPGRG; QNVLPLH and PLMLP	Both potent inhibitory effects against enzymatic markers involved in diabetes, e.g., α-amylase, α-glucosidase and DPP-IV
Rivero-Pino, et al. [19]	Soy, Lupine, and Quinoa	EPAAV, NPLL, and APFTVV	Soy the most activity but chickpea, lentil, and pea also showed potent DPP-IV inhibitory activity.
Feng, et al. [43]	Camellia seed cake (*Camellia oleífera)*	SPGYYDGR, GLTSLDRYK, and GHSLESIK	Alcalase and Asp 542 was recognized as the key target amino acid of a-glucosidase.
Gao, et al. [44]	Bovine α-lactalbumin	ELKDLKGY and ILDKVGINY	These two peptides could bind with DPP-IV.
Ibrahim, et al. [38]	Synthetic peptides	STYV; STY; SEPA; SVPA	α-glucosidase inhibitory activity: STYV > STY > SEPA > SVPA; DPP-IV: SVPA; In vitro studies: SEPA.
Jia, et al. [45]	Whey protein	LDQWLCEK, VGINYWLAHK, LDQWLCEKL, KILDKVGINYWLAHK, ILDKVGINYWLAHK	The peptide LDQWLCEKL exhibited the highest inhibitory activity.
Jin, et al. [10]	Atlantic salmon (*Salmo salar*) skin	LDKVFR	Hydrolysate with MW < 3 kDa was an excellent source of DPP-IV inhibitory peptides.
Nongonierma, et al. [46]	Camel milk (*Camelius dromedaries*)	VPV, VPF, LPVPQ, YPI, and VL	The stability of VPV to gastric and intestinal digestive enzymes suggests that it may have potential as an antidiabetic agent for humans.
Vilcacundo, et al. [47]	Kiwicha (*Amaranthus caudatus*)	FLISCLL, SVFDEELS, and DFIILE	ACE, DPP-IV, and colon cancer cell viability were obtained. These digests also showed moderate α-amylase inhibitory activity.
Wang, et al. [48]	Soy protein	LLPLPVLK; SWLRL and WLRL	Development of novel antidiabetic peptide nutraceuticals with α-glucosidase, DPP-IV, and ACE inhibitory potential.
Xu, et al. [49]	Rapeseed (*Brassica napus*) napin	PAGPF, KTMPGP, IPQVS, and ELHQEEPL	
Zheng, et al. [50]	Casein-derived synthetic peptide	VPYPQ	VPYPQ was a promising casein-derived DPP-IV inhibitor.
Ibrahim, et al. [51]	Synthetic peptides	SVPA and SEPA	Two novel and active α-glucosidase inhibitory peptides were identified; they could resist GIT digestion and have the potential to retard postprandial hyperglycemia in diabetic patients.
Mune, et al. [15]	*Bambara bean*	IP, LN, VE, and VY	After simulated digestion, thermolysin showed significantly higher ACE and DPP-IV inhibitory properties compared to the Alcalase.
Nongonierma, et al. [39]	Camel whey protein (*Camelus dromedarius*)	FLQY, FQLGASPY, ILDKEGIDY, ILELA, LLQLEAIR, LPVP, LQALHQGQIV, MPVQA, and SPVVPF	LPVP and MPVQA, with DPP-IV inhibition, were identified for the first time in camel milk protein hydrolysates.
Ji, et al. [29]	Antarctic krill (*Euphausia superba)*	AP and IPA	Can be considered as a promising source of DPP-IV inhibitory peptides for use as natural food ingredients against type 2 diabetes.
Ji, et al. [30]	Antarctic krill(*Euphausia superba)*	LVGPLP and PAL	These peptides exhibited dual inhibition of ACE and DPP-IV.
Liu, et al. [52]	*Ruditapes philippinarum* hydrolysate	LAPSTM	*R. philippinarum*-derived peptides may have potential as functional food ingredients for prevention of diabetes.
Mojica, et al. [36]	Common bean (*Phaseolus vulgaris* L.)	KKSSG, KTYGL, GGGLHK, and CPGNK	The first report. Significant antioxidant, antidiabetic, and antihypertensive properties were found after gastrointestinal simulated digestion, and inhibition of DPP-IV and α-glucosidase.
Taga, et al. [53]	Wheat gluten	GPG, QPQ, QPF, LPQ, and SPQ	The novel gluten hydrolysate prepared using ginger protease can be used as functional food for patients with type 2 diabetes.
Uraipong and Zhao [41]	Rice bran (cultivar Reiziq)	GE, GG, GP, EK, and GH	*In vitro* simulated human gastrointestinal digestion led to substantial hydrolysis of these proteins, and the resultant peptides possessed significant -glucosidase and ACE inhibitory activities.
Vilcacundo, et al. [26]	Quinoa (*Chenopodium quinoa* Willd.)	IQAEGGLT, DKDYPK, and GEHGSDGNV	The peptides generated showed ability to inhibit enzymes involved in incretin degradation and digestion of dietary carbohydrates.
Lammi, et al. [54]	Soy and Lupin Protein	Soy 1 (IAVPTGVA) and Lup 1 (LTFPGSAED)	Soy 1 (IAVPTGVA), Soy 2 (YVVNPDNDEN), Soy 3 (YVVNPDNNEN), Lup 1 (LTFPGSAED), Lup 2 (LILPKHSDAD), and Lup 3 (GQEQSHQDEGVIVR), were screened for their capacity to inhibit the activity of DPP-IV, using an in vitro bioassay against human recombinant DPP-IV.
Nongonierma, et al. [55]	Bovine α-lactalbumin	GY, GL, GI, NY, and WL	This preliminary study demonstrated the benefit of using a targeted approach combined with an experimental design for generation of dietary protein hydrolysates with DPP-IV inhibitory properties.

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
