# Peer review of "Critical Review for the Production of Antidiabetic Peptides by a Bibliometric Approach"

_nutrients, 2022, doi:10.3390/nu14204275_

Round 1
Reviewer 1 Report
The bibliographic review on antiadiabetic peptides is very interesting and complete. the study was conducted on recent articles and using suitable reference sources. In any case some small inaccuracies in the results and discussion must be corrected.
Paragraph from lines 107 to 115 should be moved and contextualized in the introduction. line 142: change the font and position of the caption.
lines 154-157 and 162-172: enlarge the text font.
Line 173: remove the first phase because it has already been repeated many times.
Line 182: change the font and position of the caption.
Line 409: change the font and position of the caption.
Tables: improve the appearance of the tables in particular that the author is readable and move each table to the point in the text where it is cited.
Author Response
Journal - Nutrients (ISSN 2072-6643)
Manuscript ID - nutrients-1871606
Type - Review
Title - Antidiabetic peptides production: a bibliometric review
New title: Critical review for the production of antidiabetic pep-tides by a bibliometric approach
Authors - Ticiane Carvalho Farias, Thaiza S. P. de Souza, Ana Elizabeth Cavalcante Fai, Maria Gabriela Bello Koblitz*
Section - Proteins and Amino Acids
Collection - Bioactive Peptides: Challenges and Opportunities
Response to Reviewer #1
The bibliographic review on antidiabetic peptides is very interesting and complete. the study was conducted on recent articles and using suitable reference sources. In any case some small inaccuracies in the results and discussion must be corrected.
We thank the reviewer for his/her thorough evaluation of the manuscript and for the constructive criticisms. We endeavored to improve the manuscript in light of his/her suggestions. Point-by-point responses to comments follow bellow.
- Paragraph from lines 107 to 115 should be moved and contextualized in the introduction
Done. Please refer to the altered manuscript (R1) lines 54 to 64.
- line 142: change the font and position of the caption.
At the suggestion of another reviewer, this figure and its caption were removed from the manuscript.
- lines 154-157 and 162-172: enlarge the text font.
Done. Please refer to the altered manuscript (R1) lines 180 to 183 and 187 to 195.
- Line 173: remove the first phase because it has already been repeated many times.
Done. Thank you for pointing that out.
- Line 182: change the font and position of the caption.
Done. All captions were formatted with the same font and in the same position
- Line 409: change the font and position of the caption.
Done. All captions were formatted with the same font and in the same position
- Tables: improve the appearance of the tables in particular that the author is readable and move each table to the point in the text where it is cited.
Done. Thank you for pointing this glitch out. Please refer to table 1 in the altered manuscript (R1).

Reviewer 2 Report
The idea of the manuscript is good, since in the last ten years, a lot of manuscripts dealing with biological activity of bioactive peptides are published, making bibliographic research difficult. The idea of the manuscript is good, since in the last ten years, a lot of manuscripts dealing with biological activity of bioactive peptides are published, making bibliographic research difficult. However, the manuscript lacks continuity (the first sections are nerely bibliometrics, the last sections like traditional review) and a deep revision is needed.
General considerations
A mere descriptive bibliometric review is not very useful and the first part of the manuscript (3.1 and 3.2 sections) should be written much more succinctly. Figures 2 and 3 also do not expand the knowledge and could be eliminated and replaced with a single simple table.
On the contrary, the final sections of the manuscript (3.5 and 3.6) as well as Table 1 can be of great interest and they could be improved. As a general suggestion, a Network visualization map, like this used in figure 5, but referring to the biological activity of the peptides identified so far, may be very useful to help the researchers to unravel the biological activities of the most studied peptides.
Finally, a different title would be preferable: a bibliometric review
Specific topics to be considered
Line 60: by virtue of their natural origin: Natural origin is not synonymous with non-toxicity. Please, rewrite the sentence
Page 3, Figure 1: screening section: it is not clear the exclusion process. Please add a sentence to explain how the most of the identified articles have been excluded. It is right the exclusion of non-research papers, but what is the meaning of type and author missing?
Line 129: Among the 75 papers, the authors included 11 reviews: in my opinion, reviews are not to be considered research work. It is another reason to better explain the screening section of the Figure 1.
Line 180: General consideration: 3.1 results section is a simple description of the type, origin and year of publication of the selected papers. I do not think it can be of interest for a researcher, at least in this, very detailed form.
Lines 223-225: the sentence is not clear. More generally, I think that, in a work of this kind, a greater distinction should be made between in vitro and in vivo methods. Some in vitro methods are referred to throughout the manuscript, but there is no mention of any experimental work on cell cultures and model organisms. Only one work on mice is considered in the text. A broader description of the various methods used to determine the biological activity of bioactive peptides should be included
Lines 401-403: The sentences are not clear. Are they to be completed?
Author Response
Journal - Nutrients (ISSN 2072-6643)
Manuscript ID - nutrients-1871606
Type - Review
Title - Antidiabetic peptides production: a bibliometric review
New title - Critical review for the production of antidiabetic peptides by a bibliometric approach
Authors - Ticiane Carvalho Farias, Thaiza S. P. de Souza, Ana Elizabeth Cavalcante Fai, Maria Gabriela Bello Koblitz*
Section - Proteins and Amino Acids
Collection - Bioactive Peptides: Challenges and Opportunities
Response to Reviewer #2
The idea of the manuscript is good, since in the last ten years, a lot of manuscripts dealing with biological activity of bioactive peptides are published, making bibliographic research difficult. However, the manuscript lacks continuity (the first sections are merely bibliometrics, the last sections like traditional review) and a deep revision is needed.
We thank the reviewer for his/her thorough evaluation of the manuscript and for the constructive criticisms. We endeavored to alter the manuscript in light of his/her suggestions, and we believe that the final result turned out much improved. Point-by-point responses to comments follow bellow.
General considerations
- A mere descriptive bibliometric review is not very useful, and the first part of the manuscript (3.1 and 3.2 sections) should be written much more succinctly. Figures 2 and 3 also do not expand the knowledge and could be eliminated and replaced with a single simple table.
Thanks for the suggestion, we completely agree with the reviewer’s appraisal. The sections 3.1 and 3.2 were significantly reduced from 1336 to 763 words, and the Figures 2 and 3 were eliminated. Supplementary table 2 presents information about the authors' names, paper's year of publication, and their respective countries.
- On the contrary, the final sections of the manuscript (3.5 and 3.6) as well as Table 1 can be of great interest, and they could be improved. As a general suggestion, a Network visualization map, like this used in figure 5, but referring to the biological activity of the peptides identified so far, may be very useful to help the researchers to unravel the biological activities of the most studied peptides.
Thanks for the suggestion, we agree again with the reviewer's assessment. A network visualization map was used to replace the Figure 7, now Figure 5 – Network visualization map of the studied bioactive peptides with multifunctional activities (VOSviewer).
- Finally, a different title would be preferable: a bibliometric review
We thank the reviewer for pointing that out. We suggested a new title: "Critical review for the production of antidiabetic peptides by a bibliometric approach"
Specific topics to be considered
- Line 60: by virtue of their natural origin: Natural origin is not synonymous with non-toxicity. Please, rewrite the sentence
Done. Thank you for pointing that out. The sentence was rewritten. Please refer to the altered manuscript (R1) line 77.
- Page 3, Figure 1: screening section: it is not clear the exclusion process. Please add a sentence to explain how most of the identified articles have been excluded. It is right the exclusion of non-research papers, but what is the meaning of type and author missing?
Thank you for raising this issue. The methodology section has been subdivided into 3 parts to better explain the process of searching, selecting and eliminating the works to be included in the review. Please refer to the altered manuscript (R1) lines 104 to 147. As a result, we believe that Figure 1 is now much clearly explained.
- Line 129: Among the 75 papers, the authors included 11 reviews: in my opinion, reviews are not to be considered research work. It is another reason to better explain the screening section of the Figure 1.
Thank you for raising this issue. Bibliometric reviews are relatively infrequent works, at least in the area of food science. Thus, the methodology involved in their elaboration is not particularly clear. This work was based on the methodology proposed by Randhawa et al. (2016), a bibliometric review in the area of innovation management. In their work, the authors used a total of 24 review articles to compose their study. A more recent reference that deals specifically with the challenges of composing bibliometric reviews, the work by Romanelli et al. (2021), argues that bibliometrics can provide important research insights by providing summarized and methodologically reliable information for various purposes. In addition to pragmatically identifying and quantifying the articles generated from empirical data, they also have the function of demonstrating interest in specific topics in a given period, and in this regard, literature reviews are included. Based on the reflections identified in these review articles, it was also possible to deepen the discussion of the study. For these reasons, we kept these reviews in the inclusion criteria of the search for the ongoing studies. Bibliometrics, including empirical data articles and literature reviews, can fundamentally contribute to this purpose by making it possible to examine how specific themes are developing and how they connect, revealing the structure of lines of research comprehensively.
*Romanelli et al., 2021. Four challenges when conducting bibliometric reviews and how to deal with them. Environmental Science and Pollution Research, https://doi.org/10.1007/s11356-021-16420-x
- Line 180: General consideration: 3.1 results section is a simple description of the type, origin and year of publication of the selected papers. I do not think it can be of interest for a researcher, at least in this, very detailed form.
Thanks for the suggestion. We agree with your consideration and the section 3.1 is presented in a more concise way, with less details.
- Lines 223-225: the sentence is not clear. More generally, I think that, in a work of this kind, a greater distinction should be made between in vitro and in vivo methods. Some in vitro methods are referred to throughout the manuscript, but there is no mention of any experimental work on cell cultures and model organisms. Only one work on mice is considered in the text. A broader description of the various methods used to determine the biological activity of bioactive peptides should be included
We thank the reviewer for the insight. We completely agree that a more thorough assessment of the works on in vitro studies was necessary and have included a couple of paragraphs on the subject. Please refer to the altered manuscript (R1) lines 604-638.
- Lines 401-403: The sentences are not clear. Are they to be completed?
Thank you for pointing that out. The sentences were rewritten. Please refer to the altered manuscript (R1) lines 423 to 426.

Round 2
Reviewer 2 Report
Dear Authors,
I checked the revised version and I was pleased to see that you have made all the changes I suggested.
I think that the manuscript can be suitable for publication in the present form.